# A well-timed shift from local to global agreements accelerates climate change mitigation

Vadim A. Karatayev[1✉], Vítor V. Vasconcelos [2,3,4,5,6], Anne-Sophie Lafuite[1], Simon A. Levin [4,6,7,8], Chris T. Bauch [9] & Madhur Anand[1]

Recent attempts at cooperating on climate change mitigation highlight the limited efficacy of large-scale negotiations, when commitment to mitigation is costly and initially rare. Deepening existing voluntary mitigation pledges could require more stringent, legally-binding agreements that currently remain untenable at the global scale. Building-blocks approaches promise greater success by localizing agreements to regions or few-nation summits, but risk slowing mitigation adoption globally. Here, we show that a well-timed policy shift from local to global legally-binding agreements can dramatically accelerate mitigation compared to using only local, only global, or both agreement types simultaneously. This highlights the scale-specific roles of mitigation incentives: local agreements promote and sustain mitigation commitments in early-adopting groups, after which global agreements rapidly draw in late-adopting groups. We conclude that focusing negotiations on local legally-binding agreements and, as these become common, a renewed pursuit of stringent, legally-binding world-wide agreements could best overcome many current challenges facing climate mitigation.

[1] School of Environmental Sciences, University of Guelph, Guelph, ON, Canada. [2] Informatics Institute and Institute for Advanced Study, University of Amsterdam, Amsterdam, The Netherlands. [3] Princeton Institute for International and Regional Studies, Princeton University, Princeton, NJ, USA. [4] Department of Ecology and Evolutionary Biology, Princeton University, Princeton, NJ, USA. [5] Andlinger Center for Energy and the Environment, Princeton University, Princeton, NJ, USA. [6] High Meadows Environmental Institute, Princeton University, Princeton, NJ, USA. [7] Resources for the Future, Washington, DC, USA. [8] Beijer Institute of Ecological Economics, Stockholm, Sweden. [9] Department of Applied Mathematics, University of Waterloo, Waterloo, ON, Canada. ✉email: vkaratay@uoguelph.ca

Existing attempts to realize international cooperation on halting climate change highlight that many ecosystem services are public goods that incentivize beneficiaries to free-ride on others' mitigation efforts[1]. The limited capacity of the atmosphere to absorb greenhouse gases before emissions severely affect climate globally is one public good that can be preserved though investments to reduce emissions ("mitigation" hereafter)[2,3]. Adopting alternative investment opportunities with faster payoffs, however, can delay cooperation unless it is reinforced by social norms[4] or short-term climate forecasts become severe[5]. Players can also delay mitigation to wait for others to develop mitigation technology and infrastructure, or to wait for rival players to invest first and give up their competitive advantage[6]. At the same time, achieving climate change mitigation globally and rapidly is critical to minimizing adaptation costs, and in preventing irreversible ecosystem degradation (e.g., species loss[7,8]), degradation of human well-being, especially in developing countries that have lower capacity for climate adaptation[9,10], and preempting the collapse of favorable climate regimes when human impacts exceed environmental tipping points[11,12].

A best-case goal of climate negotiations is a legally binding global agreement targeting < 2 °C warming by 2100 and incorporates sanctions such as trade tariffs against free-riding states that do not meet this target ("global agreements" hereafter). The failure of Kyoto, Copenhagen, and Doha negotiations to establish and draw high participation in such an agreement[13] underscore the difficulty of establishing cooperation when most players do not contribute and oppose institutions that would sanction them, leaving few mechanisms to prevent free-riding[6]. Subsequent bilateral (e.g., US-China) and pledge and review (Paris) agreements have helped accelerate emissions reductions that are inexpensive or have secondary benefits (e.g., reducing pollution[13]), but focus on voluntary commitments that may be too weak to ensure against severe climate change (>2 °C warming[12,14]). This raises the issue of what policy strategies can establish high participation in stronger mitigation commitments, backed by enforceable sanctions or tariffs[15], to complement existing agreements.

The failure of negotiations toward a global agreement that fully addresses climate change has led to a focus on building-block approaches where global cooperation grows from local agreements[1,16,17]. One model for localized agreements is built on climate clubs, where coalitions of actively interacting states (e.g., European Union, US Climate Alliance) expand existing trade or political agreements to incorporate sanctions such as trade tariffs on non-mitigating players[18,19]. To limit the potential scope and cost of sanctions[20], recent building-block work additionally focuses sanctioning interactions within sub-groups of players[21], for instance an E.U. climate agreement that sanctions only non-compliant E.U. countries. Game-theoretic[21–25] and common pool resource[26,27] work predicts that such localized mitigation agreements might succeed where global cooperation fails by reducing the number of interacting players and the potential for free-riding. However, the greater potential for bottom-up approaches to achieve mitigation locally compared with global agreements can trade off with a much longer time to achieving mitigation globally because (1) multiple groups must each adopt a climate mitigation stance[28] and (2) groups committed to mitigation have few mechanisms to influence outside late-adopting players who do not yet mitigate. So, can societies enhance both the probability and pace of desired outcomes, such as reducing greenhouse gas emissions, by combining stringent and enforceable mitigation agreements at both local and global scales?

## Results

To answer this question, we consider a stochastic system where players belong to $L$ groups, representing coalitions of countries with shared interests or geography (e.g., BRICS, AILAC), summits between trading partners (e.g., Europe-China), formal unions (e.g., EU), or countries with partly independent states (e.g., US Climate Alliance). These groups may overlap through shared members and differ in size, for instance when the biggest emitters partake in exclusive negotiations. In each group, we distinguish players that make mild emissions reductions in line with a > 2 °C warming scenario from those who adopt more stringent mitigation policies following a ≤ 2 °C scenario ("mitigators" hereafter). Which strategy players adopt is occasionally ($\mu = 5\%$ of decisions) random as chance events dominate policy, for instance when climate disasters favor mitigation or energy shortages or fiscal crises favor non-mitigation. Most of the time, however, players discern the costs of each strategy to a degree controlled by the parameter $\beta$ and choose the strategy with the greater payoff.

For many decision-makers, long-term payoffs (i.e., avoided costs) associated with global cooperation on climate do not yet underpin national commitments, as opposed to mitigation costs as alternative investments yield faster profits[29], for instance subsidizing education instead of replacing recently built coal plants. Hence, we focus on a worst-case negotiations scenario where the costs of stringent mitigation substantially outweigh both the benefits of reducing climate change (e.g., global food security or ameliorating climate disasters[19]) and secondary, group-level mitigation benefits (e.g., reduced air pollution), for a net mitigation cost $c$ (see Methods).

Mitigators support legally binding agreements that, when sufficiently endorsed, punish players with minimal mitigation commitments trough fines or trade tariffs. Such agreements can be local, affecting players only within a specific group, global, supported by and affecting all players across groups, or both[30]. This structure is analogous to sanction-based climate clubs[18] but where (1) for local agreements, clubs are localized and only admit new members (and punish non-members) who belong to the same group or agreement; and (2) club members (mitigators) punish non-members (non-mitigators) by voting for sanctions or trade tariffs within existing local (group-level, e.g., E.U.) or global institutions (e.g., W.T.O.). We also account for the fact that mitigators can incur sanctioning costs in active agreements, for instance when non-mitigating states reciprocate trade tariffs. Starting from a base model and then expanding to heterogeneous costs, players influence and emissions, group size, and group rivalry, we examine whether bottom-up (local) agreements, world-wide (global) agreements, or a combination of these two best promotes fast climate change mitigation.

**System dynamics**. As observed in theory[21,31] and reality[32], across replicate simulations we find that premature global-scale efforts fail to accelerate cooperation on climate (Fig. 1a). Although exogenous chance events occasionally produce brief mitigation efforts within individual groups, globally mitigation never reaches the quorum required to institute a global legally binding agreement (Fig. 1b.iii). In contrast, local legally binding agreements can capitalize on stochastically induced, mitigator dominance within a group to establish mitigation as the local norm. Eventually, this process occurs in every local group, translating to global climate mitigation (Fig. 1b.ii). Next, we consider a hybrid agreement strategy that shifts emphasis from local to global agreements as global mitigation increases and later compare the case where states attempt local and global agreements concurrently.

Once local agreements produce a sufficiently high global proportion of mitigators, a shift to global agreements more than doubles the rate at which players commit to mitigation

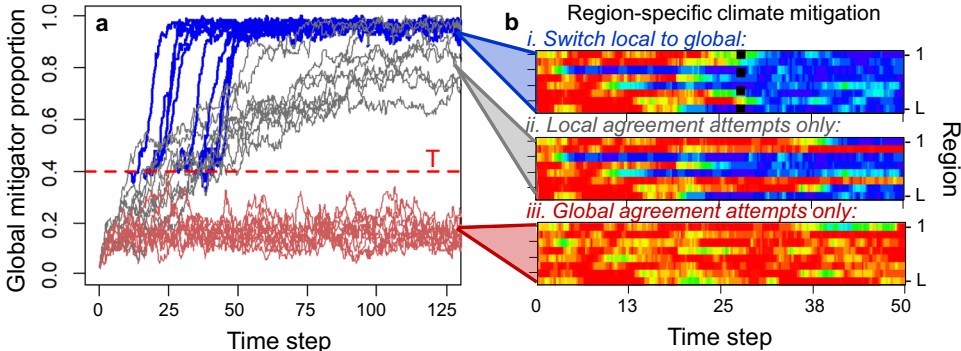

**Fig. 1 Mitigation is best achieved through a timely shift from local to global agreements, on account of scale-specific mechanisms. a** Time series of the proportion of system-wide climate mitigation over ten replicate model runs with global agreements only (red), local agreements only (gray), or a shift from local to global agreements (blue lines) once mitigation is sufficiently common (threshold $T = 0.4$). **b** Proportions of mitigators from 0 (red) to 100% (blue) across regions 1 to $L$ (y-axes) and time (x-axes) within a single model run under three approaches (i–iii). Unlike global agreements (**b**.iii), local agreements are critical for establishing mitigation as the norm in early-adopting regions (e.g., region 3 in **b**.ii); global agreements spread mitigation across regions once it is sufficiently common (**b**.i, dashed line denoting shift to global at $T$). These simulations model equal emissions across players, so that mitigation proportions equal the proportion of all players adopting a stringent mitigation strategy. Note that for global agreements only, mitigator proportions never reach the quorum $p_T = 0.5$ required to enact agreements and non-mitigators are never sanctioned. Simulations show eight regions with 18 players in each and no among-region rivalry or economies of scale; all other parameters values listed in Supplementary Fig. 1.

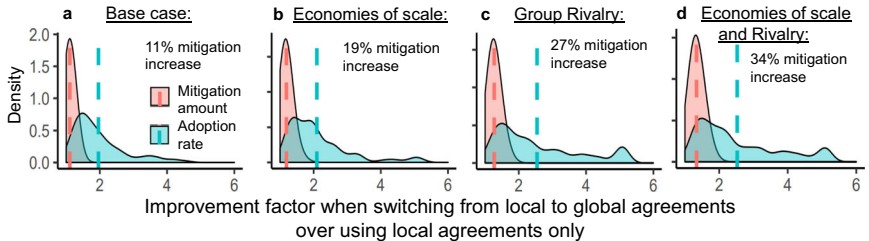

**Fig. 2 The benefit of shifting from local to global agreements increases with economies of scale and group rivalry.** Panels show the factor by which shifting outperforms local-only approaches (i.e., $2 = 200\%$ increase) in mitigation adoption rate (time until 80% of players mitigate, teal) and total amount of mitigation achieved (by time step 625, red) in simulations of **a** the base model, **b** the model with economies of scale, **c** the model with group rivalry, and **d** both economies of scale and rivalry. Distributions give relative improvements in rates and total mitigation found within each of 100 replicate model runs, and vertical dashed lines denote mean improvement in each metric. Note that shifting outperformed local-only strategies in all cases (i.e., all values > 1). Shift from local to global negotiations happens at $T = 0.4$; all other parameters values listed in Supplementary Fig. 1.

(Figs. 1b and 2a). This difference arises because sanctions in global agreements push non-mitigating groups to cooperate on reducing emissions. The benefit of shifting to global agreements increases with mitigation cost, sanctioning cost, and the extent to which policy-makers discern and select the strategy with the greater payoff (see Supplementary Fig. 1 for sensitivity analysis). Each of these aspects slows the adoption of local agreements (and, accordingly, global mitigation through only local agreements) because payoff-based policy choice strongly favors non-mitigation in the absence of legally binding agreements. Notably, the threshold required for a shift to be most effective corresponds to the 50% quorum needed to enact global agreements (Fig. 1a), while delayed transitions squander the benefits of global agreements (Supplementary Fig. 2).

**Overcoming obstacles to mitigation.** Bootstrapping cooperation through local agreements may face difficulties. Investments in a shared technology, infrastructure, or public support create impediments to early motivations. Typically, these investments are initially more costly and become progressively more accessible only as they are implemented, as more nearby players become mitigators. This occurs through research and development of industries of trading partners and industrially related products[33],

and economies of scale (we refer to these effects collectively as economies of scale). When there are substantial gains for scaling, even though initial costs are high, we observe that such economies of scale increase the benefits of shifting to global agreements (19%, Fig. 2b) as inter-group interactions pull along non-mitigating groups where mitigation costs remain high.

A second major obstacle in climate change cooperation arises at the global scale from game-of-chicken dynamics among economically competing regions, for which investment in mitigation can reduce the competitiveness of regional economies through opportunity costs[6]. For instance, China maintains that industrialized countries should invest in mitigation first so that other countries can catch up economically, while industrialized countries such as the United States wish to retain a competitive advantage. We find that such inter-group rivalry disproportionally slows down mitigation through locally based agreements. As local agreements provide no incentive for pairs of intensely competing regions to commit to mitigation, a shift to global institutions is more critical in pulling along holdout regions (Fig. 2c). Unlike economies of scale, group rivalry is a mechanism holding back mitigation that cannot be solved by local agreements: a group that fully commits to mitigation and overcomes local economies of scale still faces high costs from a rival group that does not adopt mitigation. This distinction between local (economies of scale) and global (rivalry) obstacles

to mitigation is highlighted by the fact that, with local agreements only, some regions experiencing strong competition never adopt mitigation.

**Pursuing local and global agreements together or in sequence.** Given that both local and global agreements are critical to promoting mitigation, does a concerted strategy attempting agreements at both scales simultaneously improve outcomes or does it risk diluting limited resources? Attempts to form legally binding agreements can involve constitutive costs—resources spent even if agreements fail to establish. Decision-makers can face limited political capital[34], for instance by choosing either a local or a global agreement as their central platform goal. Likewise when enacting timely and effective laws is not possible, sanctioning might initially depend on a few audacious players that refuse cooperation with non-mitigators at substantial cost to themselves, even if sanctions ultimately fail to promote mitigation. For instance, when aggressive mitigation commitment is rare globally, trade sanctions by Western European nations against the majority of (non-mitigating) countries might have a greater per-country negative impact on mitigating Western European nations than on non-mitigating nations.

We find that under high constitutive sanctioning costs mitigators must first focus all efforts locally, for instance refusing trade only with non-mitigators who are in their group (Fig. 3). Thereafter, we find that the shift from local to global mitigation incentives is best done earlier (i.e., lower shifting threshold $T$) when agreements are easier to establish (i.e., require less support or smaller quorums to effectively sanction defectors) or global agreements have greater efficacy (Supplementary Fig. 1), but premature shifts to global agreements can delay mitigation at low levels for extended periods (Fig. 1b.iii). For low constitutive sanctioning costs, efforts to establish local and global legally binding agreements simultaneously do little to affect outcomes (Fig. 3) because global agreements initially fail and local agreements eventually become redundant. However, a more graduated shift where local and global agreement attempts overlap might be more beneficial when early global dialog establishes a framework for later negotiations, when sanctions are more effective among neighbors than among regions, or when uncertainties exist in players' commitments to mitigation or in the quorum needed to establish a global agreement.

**Inequality in emissions and diplomacy.** A defining aspect of climate negotiations is that world powers account for a disproportionate amount of emissions and diplomatic influence[6,35]. We explore this scenario by accounting for (1) the current distribution of emissions across countries in evaluating net mitigation achieved and (2) following[35], the proportionally greater weight of higher emitters in determining whether legally binding agreements are enacted in our base model (i.e., no economies of scale or rivalry). Intuitively, player differences in emissions make mitigation outcomes more variable as the biggest emitters are early-adopters in some realizations and late-adopters in others. Diplomatic inequality however also advances the adoption of both local and global agreements, which can now arise either when a majority of low-emitters *or* a few high-emitters commit to mitigation (Fig. 4a). Global agreements are reached even faster when the biggest emitters participate in few-player groups (e.g., China-USA summits; Fig. 4b) because mitigation by the main diplomatic powers pulls the rest of the world along, although this benefit might subside or reverse if these players are also locked in economic rivalry[6]. However, we point out that shifting from local to global agreements (or, for low constitutive costs, pursuing both agreement types simultaneously) in all scenarios yields the fastest path to world-wide mitigation.

**Discussion**

We find that a timely shift from local to global legally binding agreements consistently accelerates and establishes global climate mitigation over agreements focused on a single scale. Specifically, a policy emphasis on localized legally binding agreements and, as these become more common, a renewed pursuit of a stringent and legally binding global agreement consistently accelerates climate mitigation compared with pursuing only local or only global agreements. Building on analyses of cooperation dynamics near steady-state[21], our focus on transient dynamics highlights that once they can be enacted, global agreements can play a key role by forcing emission reductions in groups, which are late to increase mitigation. This role increases as economies of scale, among-group rivalry, or high costs (Supplementary Fig. 1) make mitigation less likely to arise through chance events such as popular movements or natural disasters.

Our multi-scale model also underscores the importance of local agreements in establishing mitigation during the early phases of conservation efforts. First, local agreements can capitalize on increases in mitigation that arise from local variation but have little impact on mitigation at the global scale, as seen in lower benefits of shifting as stochasticity becomes more prevalent in player climate policy choice (see SM for description of the mitigation behavior for varying group sizes, rate of exogenous events and impact of payoff). This aspect can be especially critical as climate disasters create strong but localized momenta for climate mitigation[36,37]. Second, when favorable chance events increase mitigation in early-adopting groups (i.e., before stringent global agreements establish), local agreements "lock-in" and safeguard mitigation against subsequent chance events that may dis-favor mitigation. As an example, recent Australian wildfires reiterated climate change concerns globally, but they could create an especially strong case for changing the domestic policies in Australia towards mitigation. Additionally joining a legally binding coalition that pursues ambitious emissions reductions could then underpin continued mitigation by the member nations during a future energy crisis that might otherwise spur the construction of coal power plants. In this way, local agreements "ratchet up" mitigation globally when it is initially rare whereas global agreements fail[21].

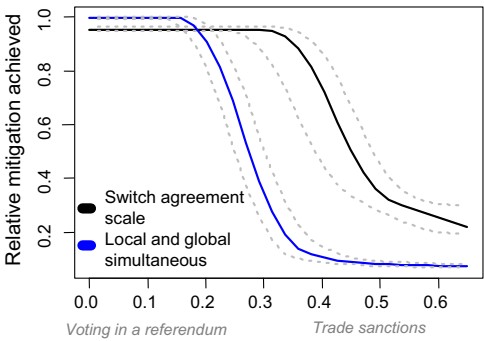

**Fig. 3 Shifting agreement scale (black, *b* = 0) outperforms simultaneous local and global agreement attempts to promote mitigation (blue, *b* = 0.5) above low constitutive agreement costs *f*.** *f* denotes the fraction of mitigator costs $p_M$ incurred at all times, regardless of whether sanctioning agreements are in effect. Solid lines denote means and dotted lines denote the 25th and 75th quantiles in mitigation achieved by time step 625 across 30 replicate trials. For *b* = 0, shift from local to global negotiations happens at *T* = 0.4; all other parameters values listed in Supplementary Fig. 1.

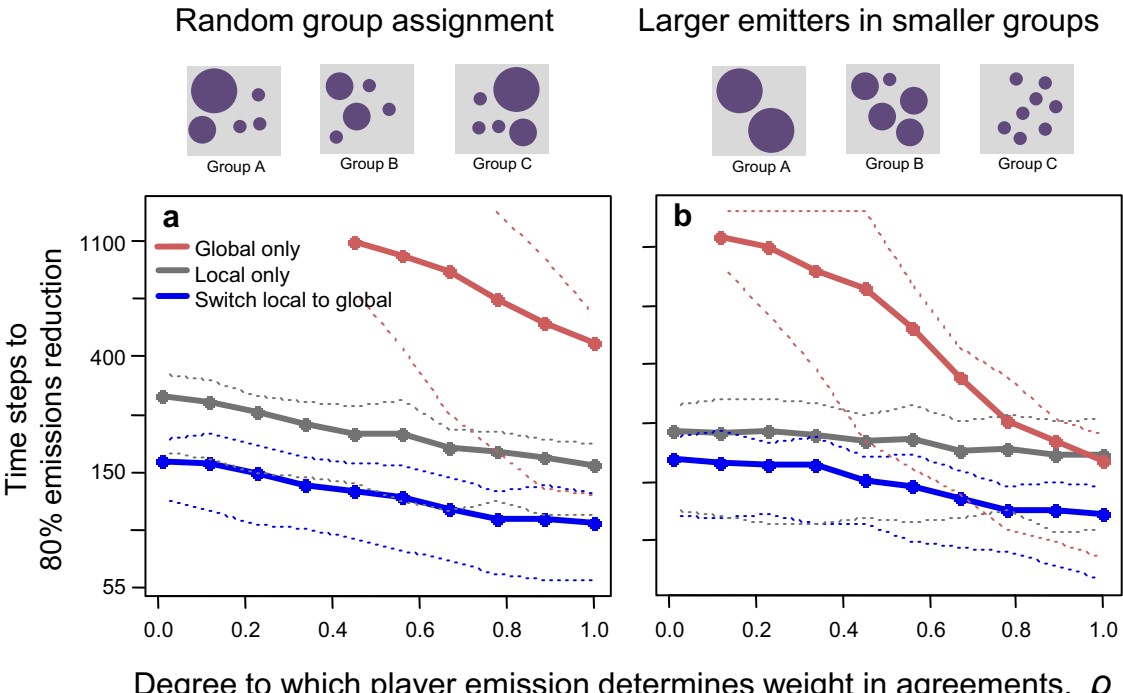

**Fig. 4 Greater weight of larger emitters in establishing enforcement-based agreements accelerates world-wide mitigation.** This pattern arises both when larger emitters are assigned to groups randomly (**a**) and when larger emitters belong to smaller groups (**b**). Plots compare the time steps to 80% world-wide mitigation reduction when using only global agreements (red), only local agreements (gray), and shifting from local to global agreements once the fraction of emissions mitigated passes $T = 0.4$ (blue). Panels above plots exemplify differences in player assignment to groups. Solid lines denote means and dotted lines denote the 25th and 75th quantiles across 100 replicate trials; note the log scale y-axis. Mitigation rates are not plotted for global-only agreements at low $\rho$ levels because this strategy failed to produce 80% world-wide mitigation in a majority of trials before time step 1500.

Our emphasis on concentrating mitigation incentives locally, followed or complemented by global-scale incentives, extends beyond the fine-based incentives examined here. First, punishment could be de-centralized and depend on individual actions, increasing proportionally with the number of mitigators willing to enact tariffs on non-mitigators (as in the climate clubs model[18]). Dynamics in such systems are similar to those presented here[21]: show that global cooperation can arise when mitigators punish only non-mitigators in their own group but not when mitigators punish all non-mitigators world-wide and face high punishment costs. Punishment can also complement positive mitigation incentives in the International Environmental Agreements framework[38], where multiple local agreements promote cooperation by reducing free-riding[23–25].

Once local efforts make stringent mitigation commitments more common, there are also several alternative strategies to rapidly expand mitigation to all other players. One strategy lies in expanding membership in agreements that control different pollutants ("regime complexes") or merging local coalitions into a global group. Likewise, overlap in group membership can accelerate mitigation globally by emphasizing the marginal gains to cooperation when information about the outcomes of action is missing[30]. Taken together, a diversity of local and global mitigation incentives could complement and reduce the extent of sanctioning required to effectively expand the voluntary mitigation commitments under current agreements.

To focus on the qualitative benefits of shifting from local to global climate agreements over alternative approaches (i.e., local, global, or simultaneous), we have omitted several features that can quantitatively affect our results and the projected pace of climate mitigation. These include the fact that mitigation commitments lie on a continuum (vs. our mitigator/non-mitigator dichotomy) as mitigation costs and benefits vary among players[13]. Following this,

the specific form of local agreements and sanctions may vary among groups from legislative (e.g., in the E.U.) to political (e.g., BRICS) or tariff-based (e.g., a USA–China negotiations). Projecting the pace and magnitude of mitigation under different scenarios may additionally require a more detailed model of how fast players reconsider and implement their climate policy (including tariffs or sanctions on non-mitigating players), how quickly climate changes under each scenario, and how climate projections influence policy decisions (e.g., ref. [5]).

Continually worsening climate will eventually force countries to reduce greenhouse gas emissions, with current debates centering on the urgency and best course of action to achieve global mitigation. We have shown one strategy that can greatly accelerate cooperation to mitigate climate change in the near future by leveraging local and global efforts.

## Methods

We consider $L \cdot N$ players that form $L$ groups around shared political, economic, or emissions reductions goals. In the main analyses we model groups that each contain $N$ randomly assigned players, and in Fig. 4 we consider the case where players with higher emissions and diplomatic weight on agreement adoption form smaller groups, such as bilateral USA–China negotiations or a G7 summit, leaving players with lower emissions in larger groups. Thus, each group $j$ contains $N_j$ players, of which $M_j$ pay a substantial net cost $c$ to reduce emissions while all others mitigate minimally, for a mitigator proportion $m_j = M_j N_j^{-1}$. Net mitigation cost $c = C - B_{\text{local}} - B_{\text{global}}$ implicitly combines (1) mitigation cost $C$, (2) the global benefit of each mitigator ameliorating climate change $B_{\text{global}}$, and (3) secondary benefits of each mitigator to their group $B_{\text{local}}$, for instance through reduced regional air pollution or a technological stimulus from renewable energy sectors. Note that this formulation omits variation in costs among players (explored in ref. [35]), increased benefits if climate change impacts become severe (explored in refs. [5,35]) and assumes benefits equally affect all players in a group (for $B_{\text{local}}$) or the world (for $B_{\text{global}}$).

When mitigators reach a majority, consensus establishes legally binding agreements that sanction non-mitigators. We consider a combination of local and global sanctioning efforts where mitigators initially attempt to form local agreements until

global mitigation proportion $m_G = (LN)^{-1}\sum_j M_j$ exceeds a threshold $T + b$, and attempt to form global agreements once global mitigation proportion exceeds a threshold $T$. In the base model, we assume negotiations gradually shift from local to global agreements as $m_G$ exceeds $T$ with $b = 0$. Alternatively, agreements may overlap to a degree $b > 0$, in which case the point at which local agreements are phased out $T + b$ occurs after global agreement attempts begin at $T - b$ and, potentially, non-mitigators may be punished by both local and global agreements. We consider four types of agreement strategies: (i) coordinated flip from local to global agreements, $0 < T < 1$, $b = 0$; (ii) purely local agreements, $T = 1$, $b = 0$; (iii) purely global agreements, $T = b = 0$; and (iv) coexistence of local and global agreement attempts at all times, $T = 0$, $b = 1$.

Throughout, we use a decreasing sigmoid function $\Theta(x) = (1 + \exp(hx))^{-1}$ to model (1) the gradual implementation of punishment agreements once mitigator proportion exceeds $p_T = 0.5$ and (2) smooth transitions from local to global agreements ($h = 20$, $0 \leq x \leq 1$). Thus, the amount of punishment attempted by mitigators is $\delta(m_j) = \Theta(m_G - T - b) + \Theta(T - m_G)$ and reflects a shift from local to global agreement attempts as $m_G$ exceeds $T$. Finally, we allow global agreements to differ from local agreements in sanctioning efficacy by a factor $E$. The amount of punishment realized by successfully established agreements is then

$$\Delta(m_j) = \Theta(m_G - T - b)\Theta(p_T - m_j) + \Theta(T - m_G)\Theta(p_T - m_G)E. \quad (1)$$

Payoffs depend on active punishment agreements (denoted by $\Delta(m_j)$), the player's mitigation policy $s$, and the cost of mitigation $c$. When active, agreements punish non-mitigators by an amount $p_{NM}$ but can also cost mitigators an amount $p_M < p_{NM}$, for instance when sanctions prohibit lucrative trade; further, mitigators incur a fraction $f$ of punishment costs even if non-mitigators are not sanctioned when sanctioning coalitions, institutions, or bills fail to establish. This yields the climate policy-specific payoffs

$$\Pi_{NM}(m_j) = -p_{NM}\Delta(m_j) \quad (2)$$

$$\Pi_M(m_j) = -c - p_M(f\delta(m_j) + (1 - f)\Delta(m_j)). \quad (3)$$

At each time step, one randomly chosen player decides to reconsider their climate policy with probability $\kappa$, the social learning rate. Players tend to choose strategies with greater payoffs to a degree $\beta$, which reflects the importance of financial considerations in climate policy. With probability $\mu$, however, chance events supersede any payoff differences and players choose strategies at random. Thus, a player with current policy $s$ in group $j$ switches to policy $k$ (with $n_{k,j}$ denoting the number of players with policy $k$) with probability given by the Fermi function

$$\Pr_{s \to k,j} = \kappa\left(\frac{(1 - \mu)n_{k,j}N_j^{-1}}{1 + \exp(\beta(\Pi_s(m_j) - \Pi_k(m_j)))} + \mu\right). \quad (4)$$

To model economies of scale, we set initial mitigation costs $c_0$ that decline by a fraction $r_L$ past $m_j = r_T$ using the updated mitigation cost $c(m_j) = c_0(1 - r_L\Theta(r_T - m_j))$, where $c_0 = c(r_T + (1 - r_T)(1 - r_L))^{-1}$ ensures that $\int_{m_j=0}^{1} c(m_j) = c$ in our base model.

To account for the effects of among-region economic competition, we model reduced payoffs for regions where players commit to climate mitigation in lieu of alternative investments to bolster their economies. Among-group interactions in the matrix $\mathbf{A}$ are 0 for $a_{i=j}$ while randomized $a_{i\neq j}$ terms are drawn from a multi-variate uniform distribution with rivalry reciprocity $R = cor(a_{i,j}, a_{j,i}) > 0$. Given group-level payoffs as $P(\vec{m}) = (\vec{m}\Pi_M(\vec{m}) + (1 - \vec{m})\Pi_{NM}(\vec{m}))(\mathbf{A} + \mathbf{I}_L)$, we normalize the rows of $\mathbf{A}$ by their sum to obtain $\mathbf{A}^N$ and arrive at the fitness of group $j$ with $n$ mitigators $\lambda_{j,n} = P(\vec{m'}) \cdot (\mathbf{A}^N - \mathbf{I}_L)_j$, where $\vec{m'}_{i=j} = n$ and $\vec{m}_i$ otherwise. We then incorporate the consequences of individual choice for group-level payoffs using the updated payoffs $\Pi_s^G(m_j) = \Pi_s(m_j) + \alpha G_s/2$, where $G_{NM} = \lambda_{j,m_j - N^{-1}} - \lambda_{j,m_j}$, $G_M = \lambda_{j,m_j + N^{-1}} - \lambda_{j,m_j}$, and $\alpha$ scales individual payoff of group competitiveness relative to punishment and mitigation costs. With this formulation mitigation adoption slows when pairs of non-mitigating regions compete intensely.

To model player differences in emissions in our base model, we assign each player $i$ a proportion of global emissions $e_i$ drawn from a lognormal distribution. Reflecting reality, we calibrate the distribution variance so that the maximum emission proportion (out of $L \cdot N = 144$ players) is, on average, 0.28 (note that this degree of inequality does not qualitatively affect our results). Next, we introduce two metrics, the fraction of emissions mitigated within each group $j$, $m_j^e$, and the fraction of emissions mitigated world-wide $m_G^e$. We model the greater diplomatic influence of higher emitters by modifying eqn. 1 so that whether agreements are enacted depends to a degree $1 - \rho$ on the proportion of mitigators and to a degree $\rho$ on the proportion of emissions mitigated:

$$\Delta(m_j) = \Theta(m_G^* - T - b)\Theta(p_T - m_j^*) + \Theta(T - m_G^*)\Theta(p_T - m_G^*)E, \quad (5)$$

where $m_G^* = (1 - \rho)m_G + \rho m_G^e$ and $m_j^* = (1 - \rho)m_j + \rho m_j^e$. Finally, we assign higher-emitting players to groups with fewer members (correlation between $e_i$ and the number of players player $i$ shares their group with $= -0.9$), which places the largest emitters in a group of two to three players and players with lower emissions into the remaining $L - 1$ groups with progressively more members.

**Reporting summary**. Further information on research design is available in the Nature Research Reporting Summary linked to this article.

## Data availability
This study does not use primary datasets.

## Code availability
This study is based only on simulation code available at ref. [39].

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

## Acknowledgements

We thank the UC Davis FARM cluster for computing resources. This research was supported by a New Frontiers in Research Fund Exploration Grant of the Government of Canada (to M.A. and C.T.B.).

## Author contributions

V.A.K. conceived the study, V.A.K., V.V.V., C.T.B. and M.A. designed and analyzed the model, and V.A.K., V.V.V., A.S.F., S.A.L., C.T.B. and M.A. participated in writing the manuscript.

## Competing interests

The authors declare no competing interests.
