## [Peer Review File · Nature Communications]

REVIEWER COMMENTS

Reviewer #1 (Remarks to the Author):

Report for NCOMMS-20-28481: "A well-timed switch from local to global agreements accelerates climate change mitigation"

This paper analyzes cooperation on climate change mitigation when countries/regions can join global and sub-global agreements. The authors compare the cases of only sub-global and only a global agreement with the possibilities of having both options in parallel and a switch from sub-global to global. They find that the latter has the potential to accelerate mitigation efforts. This is a stimulating and novel contribution to the literature that so far only analyzed scenarios with global or with sub-global agreements and compared these. It is easy to read and understand and of interest for the broad readership of Nature Communications, especially to the fields of economics of climate change and climate governance. In addition, it is relevant to policymakers. I therefore think that it is suitable for publication in Nature Communications if the authors are able to address some major and minor concerns.

Major points:

The authors should clarify the role of sanctions and the incentive structure of the model (see also comments on the Methods section below). The logic of free-rider incentives in climate change mitigation is well motivated, but from the Methods section, it is not clear which role they play in the model (see below). It seems that sanctions are crucial in the model, which is different from other contributions with multiple sub-global climate agreements (see references under minor points). Nordhaus (2015) shows how the introduction of sanctions changes the incentive structure in the formation of climate agreements. It might be useful to include a model version without sanctions and compare it to the version with sanctions to examine their role for the results.

Ecosystem restoration [line 171ff.]: To me it is not clear how the results about climate change mitigation carry over to ecosystem restoration. From the Methods part, I cannot see that ecosystems are explicitly covered in the model. If this is just a different interpretation of the model, the authors should make this transparent. The issue is: to my knowledge, restoring ecosystems might have quite some local co-benefits in addition to the mitigation effect, whereas the benefits from pure mitigation measures are mainly global (public good). This makes a crucial difference.

Methods section: Although the text is well written and easy to understand, the authors should improve/expand the Methods section. Especially the motivation of the total payoffs (equations (1) and (2)) is not clear to me. Where are the benefits from global emission reductions? I see the punishment and the mitigation costs, but the free-rider incentives are due to the benefits from mitigation (public good). Further, the authors should explicitly say that their results are based on simulations (parameters in Supplementary Figure 1).

Minor points:

Sub-global agreements do not always have to be regional (local). Coalitions might also form in line with other types of proximity than geographic (trade relations, political or religious interests etc.). This is just a suggestion.

The authors should make clear that the study is based on numerical results. Other papers provide analytical results (see below).

Line 51-53: Reference to the game-theoretic literature should be improved. Some of the cited literature is not game-theoretic and economic contributions are missing: The analytical (Asheim et al. 2006) and simulation (Osmani and Tol 2010) studies of Asheim et al. (2006) and Osmani and Tol (2010) show that two agreements can achieve a larger number of cooperating countries and higher levels of emission reductions. Hagen and Eisenack (2019) shows that the outcome can be further improved by allowing more than two sub-global agreements, but that this crucially hinges on the benefit and damage functions, because freeriding can also occur between coalitions.

Line 159-163: This sentence confuses me. Reducing emissions is a contribution to a global public good. Thus, mitigation that arises locally always has an impact at the global scale.

References:

Asheim, G. B., Froyen, C. B., Hovi, J., & Menz, F. C. (2006). Regional versus global cooperation for climate control. *Journal of Environmental Economics and Management*, 51(1), 93-109.

Hagen, A., & Eisenack, K. (2019). Climate clubs versus single coalitions: the ambition of international environmental agreements. *Climate Change Economics*, 10(03), 1950011.

Osmani, D., & Tol, R. S. (2010). The case of two self-enforcing international agreements for environmental protection with asymmetric countries. *Computational Economics*, 36(2), 93-119.

Nordhaus, W. (2015). Climate clubs: Overcoming free-riding in international climate policy. *American Economic Review*, 105(4), 1339-70.

Reviewer #2 (Remarks to the Author):

Dear authors,

I enjoyed reading this manuscript. Although I cannot assess the validity or suitability of the methodology, I found that the results on criticality of timing of the switch from local to global agreement as well as the analysis of the two critical factors (economies of scale and group rivalry) novel and interesting to the broader field of climate policy and law.

However, I have strong concerns about the relevance of the core assumptions and results and the usefulness of this work when informing international climate policy. First, a 'global agreement' is not defined, in terms of substantive or procedural criteria (eg universal country participation? legally binding mitigation targets?), nor what is considered an 'effective' global agreement (eg with mitigation targets adding up to a 2-degree target or certain carbon budget? potential or actual effect on emission reduction). Further, it is not discussed whether the current global regime (UNFCCC and Paris Agreement) meet the standard of a 'global agreement' or not, and why. This paper would be much more valuable if it is clearly related to the current, existing legally binding as well as voluntary commitments. Lacking such analysis, it comes across as highly hypothetical, and quite possibly obsolete - since many would argue that the Paris Agreement does constitute a global agreement and that the idea of legally-binding, top-down mitigation targets (a la Kyoto Protocol) is irrelevant in the current landscape. Please see work by Robert Keohane and David Victor (Cooperation and discord in global climate policy, *Nature Climate Change*, 2016) that problematises the need for formal coordination of mitigation targets vs. informal coordination of mitigation targets. For example, I wonder how the current informal negotiation between EU and China on Chinese mitigation target to be submitted as part of its updated NDC would be considered in the methodological framework of the paper?

Second, the article does not recognise situations of 'multiple externalities' (Elinor Ostrom, 2010, *Global Environmental Change*), which shows how a climate mitigation commitment may rather be motivated by local (excludable) benefits (eg improved air quality) than global benefits of regulating climate. Again, the work by Keohane and Victor, as well as much other political science scholarship, shows that the accountability relationships and the 'negotiation' may not primarily be between states, but between a government and the electorate. Of course, public interest in and influence over national climate targets differ across countries. But it is a highly constraining assumption that countries set their targets (including as and when submitted under a global agreement) only with respect to what other countries are doing, and what is the scope to free-ride.

Third, an excessively strong assumption on sequencing is made, ie that policy-makers can/should/do focus their 'efforts' on one scale at the time. This does not appear realistic, as most governments would closely coordinate what they commit under various agreements at various scales. I would also like to see the authors elaborate on the timescale, is there time to secure effective agreements (at any/all scales) with a sequencing approach in view of the how fast the 1.5 degree target window is closing?

My main concern is thus that the paper has not reached its full potential in how relevant and useful it can be, through clear references to the current legal and policy regime. I also felt that the part on ecological restoration was separate from the main argument and should rather be discussed in a separate paper, to instead use the space to elaborate on the validity of the core assumptions behind climate change emission reduction, as suggested above.

Reviewer #3 (Remarks to the Author):

This paper argues that global negotiations are slow and that regional ones can be faster but since a global participation is obviously needed, the authors argue in favour of a regime that mixes the two starting regional and at some point switching to global.

The paper makes some interesting points and is worth a read but did not fully convince me. It is clear that global climate negotiations face problems but is this the solution? There are some regions – notably the EU where there has been some progress in climate policy but there are many regions where one fails to see much progress and – worse than that one fails to see why there would be any particular compelling logic to the regional approach.

A very reasonable alternative approach is to say that you need a G2 or G3, G4 approach. The two biggest emitters are the US and China. Without them – no deal. One could argue that a reasonable approach is for these two mega-polluters and mega-powers to negotiate either bilaterally or preferably perhaps including at least the EU and India. This is the approach taken by the debate on “Climate Clubs” but the paper makes no reference to this debate nor Bill Nordhaus and others who have proposed such solutions. The paper does also not properly relate to the work by Scott Barrett on how to build coalitions and design treaties that will last and create incentives to deal with the incentive problems involved. A third approach is to break up the global treaties and deal with different pollutants, sectors or industries separately. I.e. to have one treaty for HCFCs and PFCs and one on deforestation and land use change and another on steel and cement etc.

None of this, and very little in general of the economic or social science literature is referenced or used. Instead the authors focus exclusively on the regional vs global dimension.

The argument of the paper starts by saying that “premature” global treaties don’t work. Well yes, it is true that nothing has worked yet. But is this due to lobbyism, rivalries related to other issue, a lack of understanding, faulty design, disagreement over burden sharing or what? The paper continues to argue that regional treaties will work much better. This is possible but I am not entirely convinced of the arguments. Of course if we accept this idea then it would be a good idea to start regionally and the second part follows automatically : one has to switch at some point from the regional to the global. The paper is backed up with a model about this transition that I found somewhat opaque and somewhat construed. It is quite mechanical – I don’t see the economic forces, the interests, the lobbyists. Not much effort is made to motivate model structure or parameters. Somehow I was, again, not very convinced. Are the regions equal in size? What are the real driving forces?

All in all the paper makes some good points and is interesting but I fail to be totally convinced and I also see a lack of connection to the broader literature on treaty formation. In summary I am hesitant to recommend publication at least not without a very substantial revision.

More detailed comments and questions

(p. 4, r. 83-84) How is this seen in figure 1? The variance of the local agreement is so high. It is not obvious that we are speaking of a doubling here. Please explain.

(p.4, r. 84-85) It is difficult to see in the figure what is claimed in the text that the figure shows. Doubling the speed to 80%? How does one see that in the figure?

(p.4, r. 91-92) I do not understand how one sees this in the figure. When we speak of a majority of countries – is this assuming they are identical – or do you mean a set of countries with a majority of the pollution – or of GDP?

(p. 7, r. 138-142) I don’t understand this. Could you elaborate?

(p. 10, r. 208-211) Are the thresholds really correct here? Hard to follow. If the players focus ALL their efforts on local agreements up until T_L , then what significance does T_G have? Since T_G happens before T_L ... Please explain and motivate better.

(p. 11, r. 228) More explanation needed, What is p_T ? How do we go from expression in row 226

to expression in row 228? Some more explanation would be beneficial at least in appendix or for the referee.

(p.11, r. 230) Please show how you arrive at eq 2 and eq 3?

(p.11, r. 231) How do you motivate this feature of a random switch in strategy? What real world concept does it aim to capture?

(p. 11, r 224-225) For $T=0$ and $b>1$ you say the global and local agreements overlap. But previously (p.10, r. 213) you said $T=0$ means only global agreements will take place. This is somewhat confusing.

(p.12, r. 239-240) Is strength of rivalry is randomized?

Reviewer #1

Comment: This paper analyzes cooperation on climate change mitigation when countries/regions can join global and sub-global agreements. The authors compare the cases of only sub-global and only a global agreement with the possibilities of having both options in parallel and a switch from sub-global to global. They find that the latter has the potential to accelerate mitigation efforts.

This is a stimulating and novel contribution to the literature that so far only analyzed scenarios with global or with sub-global agreements and compared these. It is easy to read and understand and of interest for the broad readership of Nature Communications, especially to the fields of economics of climate change and climate governance. In addition, it is relevant to policymakers. I therefore think that it is suitable for publication in Nature Communications if the authors are able to address some major and minor concerns.

Major points:

The authors should clarify the role of sanctions and the incentive structure of the model (see also comments on the Methods section below). The logic of free-rider incentives in climate change mitigation is well motivated, but from the Methods section, it is not clear which role they play in the model (see below). It seems that sanctions are crucial in the model, which is different from other contributions with multiple sub-global climate agreements (see references under minor points). Nordhaus (2015) shows how the introduction of sanctions changes the incentive structure in the formation of climate agreements. It might be useful to include a model version without sanctions and compare it to the version with sanctions to examine their role for the results.

Line 159-163: This sentence confuses me. Reducing emissions is a contribution to a global public good. Thus, mitigation that arises locally always has an impact at the global scale.

Response: We rewrote the introduction to clarify the kinds of agreements and mitigation commitments we focus on. Specifically, we investigate negotiation frameworks that go deeper than most current voluntary pledges (e.g., as part of Paris) by using legally-binding sanctions as part of local agreements (sensu Nordhaus climate clubs) or global agreements (sensu Kyoto), the latter of which are not currently politically feasible. The “no-sanctions” scenario is equivalent to our global agreements only simulations in Figure 1, where the 50% global quorum needed to enact a global legally-binding agreement is never reached; we clarify this in Figure 1 legend.

We also now emphasize that, while free-riding is the motivation for reducing group sizes in many deterministic models of climate cooperation, free-riding is absent in our model because we consider the present scenario where the costs of stringent mitigation outweigh its benefits in negotiations and policymaking (see response below; lines 81-89). Instead, the benefit of sub-global groups and local agreements is their ability to ‘ratchet up’ global cooperation by securing mitigation in each group as soon as a majority of players cooperate, as we now clarify in the Discussion (lines 203-219).

Comment: Ecosystem restoration [line 171ff.]: To me it is not clear how the results about climate change mitigation carry over to ecosystem restoration. From the Methods part, I cannot see that ecosystems are explicitly covered in the model. If this is just a different interpretation of the model, the authors should make this transparent. The issue is: to my knowledge, restoring ecosystems might have quite some local co-benefits in addition to the mitigation effect, whereas the benefits from pure mitigation measures are mainly global (public good). This makes a crucial difference.

Response: Thank you, we removed this section following the reviewer’s and editor’s comments.

Comment: Methods section: Although the text is well written and easy to understand, the authors should improve/expand the Methods section. Especially the motivation of the total payoffs (equations (1) and (2)) is not clear to me. Where are the benefits from global emission reductions? I see the punishment and the mitigation costs, but the free-rider incentives are due to

the benefits from mitigation (public good). Further, the authors should explicitly say that their results are based on simulations (parameters in Supplementary Figure 1).

Response: Thank you, we addressed this by better explaining the generality of our model in the Introduction and Methods. As we now clarify, our approach implicitly accounts for both regional and global benefits of mitigation, but assumes that total benefits (in the current political climate) are still outweighed by the costs of a stringent mitigation policy (i.e., in line with a <2C warming scenario) when players select policy strategies. Thus, we model a net cost c of stringent mitigation that, in the near-future, still favors very limited mitigation in line with the voluntary Paris commitments. In the Methods and Discussion we also note the limitations of this assumption over the long-term (50-100yr), for instance if severe climate change allows the benefits of mitigation to outweigh the costs of stringent mitigation in policymaking.

Minor points:

Comment: Sub-global agreements do not always have to be regional (local). Coalitions might also form in line with other types of proximity than geographic (trade relations, political or religious interests etc.). This is just a suggestion.

Response: Thank you, we now point this out and refer to “regions” as “groups” throughout. We also consider the effects of unequal group sizes where larger emitters with greater diplomatic influence participate in groups with fewer members (e.g., bilateral summits; Figure 4).

Comment: The authors should make clear that the study is based on numerical results. Other papers provide analytical results (see below).

Response: Thank you, we made the change (lines 103-104).

Comment: Line 51-53: Reference to the game-theoretic literature should be improved. Some of the cited literature is not game-theoretic and economic contributions are missing: The analytical (Asheim et al. 2006) and simulation (Osmani and Tol 2010) studies of Asheim et al. (2006) and Osmani and Tol (2010) show that two agreements can achieve a larger number of cooperating countries and higher levels of emission reductions. Hagen and Eisenack (2019) shows that the outcome can be further improved by allowing more than two sub-global agreements, but that this crucially hinges on the benefit and damage functions, because freeriding can also occur between coalitions.

Response: Thank you, we now reference this important work in the Introduction and the Discussion. In particular, we note that our results highlight a complementary benefit of multiple sub-global agreements (in addition to reducing free-riding): that given stochasticity in individual decisions, local agreements help ratchet up global mitigation by locking-in mitigation as soon as it arises in each region. We also emphasize that our results highlight the benefits of shifting

between (or coordinating) mitigation incentives at local and global scales and that, in practice, the formation and incentive structure of specific agreements can vary (lines xx-yy).

Reviewer #2

Dear authors, I enjoyed reading this manuscript. Although I cannot assess the validity or suitability of the methodology, I found that the results on criticality of timing of the switch from local to global agreement as well as the analysis of the two critical factors (economies of scale and group rivalry) novel and interesting to the broader field of climate policy and law. However, I have strong concerns about the relevance of the core assumptions and results and the usefulness of this work when informing international climate policy.

Comment: First, a 'global agreement' is not defined, in terms of substantive or procedural criteria (eg universal country participation? legally binding mitigation targets?), nor what is considered an 'effective' global agreement (eg with mitigation targets adding up to a 2-degree target or certain carbon budget? potential or actual effect on emission reduction). Further, it is not discussed whether the current global regime (UNFCCC and Paris Agreement) meet the standard of a 'global agreement' or not, and why. This paper would be much more valuable if it is clearly related to the current, existing legally binding as well as voluntary commitments. Lacking such analysis, it comes across as highly hypothetical, and quite possibly obsolete - since many would argue that the Paris Agreement does constitute a global agreement and that the idea of legally-binding, top-down mitigation targets (a la Kyoto Protocol) is irrelevant in the current landscape. Please see work by Robert Keohane and David Victor (Cooperation and discord in global climate policy, *Nature Climate Change*, 2016) that problematises the need for formal coordination of mitigation targets vs. informal coordination of mitigation targets. For example, I wonder how the current informal negotiation between EU and China on Chinese mitigation target to be submitted as part of its updated NDC would be considered in the methodological framework of the paper?

Response: Thank you for pointing this out; in the Introduction we now frame our study much more thoroughly in the context of existing agreements. In particular, we emphasize our focus is on sanction-based (legally binding) incentive strategies that can expand mitigation beyond current voluntary commitment levels under global (Paris) and bilateral agreements. Accordingly, a global agreement in our model represents stringent (e.g., <2C) mitigation commitments backed by effective sanctions a la Kyoto – an agreement which, once more countries adopt an aggressive climate mitigation strategy through a focus on sub-global agreements (e.g., EU, US Climate Initiative), could become more realistic and boost world-wide mitigation by pulling along late-adopting players. We now also consider a broader set of scenarios which account for inequality in emissions, diplomatic influence in signing such agreements, and the effects of grouping the largest emitters into few-player negotiations (e.g.,

China and EU or USA; lines 174-189, Figure 4). Finally, in the Discussion, we also emphasize that our results highlight the benefits of shifting between (or coordinating) mitigation incentives at local and global scales and that, in practice, the formation and incentive structure of specific agreements can vary (lines 220-238).

Comment: Second, the article does not recognise situations of 'multiple externalities' (Elinor Ostron, 2010, Global Environmental Change), which shows how a climate mitigation commitment may rather be motivated by local (excludable) benefits (eg improved air quality) than global benefits of regulating climate. Again, the work by Keohane and Victor, as well as much other political science scholarship, shows that the accountability relationships and the 'negotiation' may not primarily be between states, but between a government and the electorate. Of course, public interest in and influence over national climate targets differ across countries. But it is a highly constraining assumption that countries set their targets (including as and when submitted under a global agreement) only with respect to what other countries are doing, and what is the scope to free-ride.

Response: We have revised the framing and specific methods of our model to show how we implicitly account for the global benefits of climate mitigation as well as secondary, within-group benefits (e.g., reduced pollution). This better highlights that both benefits can be large but do not presently outweigh the costs of stringent mitigation in line with a <2C warming scenario; for this reason, the scope to free-ride plays a minor role in our model and in present negotiations. The revised framing of our study also better addresses the issue of accountability/negotiation: given the high net costs of stringent mitigation, we focus on what additional international policies (e.g., tariffs on exports) could expand the mitigation commitments realized under recent treaties – i.e., emissions reductions which were already economically sensible due to secondary benefits or improved technologies (line xx-yy). Finally, in an added Discussion paragraphs on model assumptions, we highlight how our results could quantitatively depend on the fact that realized climate policies span a continuum of mitigation levels.

Comment: Third, an excessively strong assumption on sequencing is made, ie that policy-makers can/should/do focus their 'efforts' on one scale at the time. This does not appear realistic, as most governments would closely coordinate what they commit under various agreements at various scales.

Response: In the revised Introduction, we now focus on motivating a “multifaceted strategy” and then step through switching first followed by a case with simultaneous agreements. We also reiterate in the Discussion that the proposed approach here does not exclude existing voluntary global agreements, but rather can generate cumulative benefits with simultaneous coordination at all scales. To account for this, we now refer to “shifting” rather than “switching” from local to global agreements throughout (including the title) to reduce the emphasis on an “either-or”

idea. Finally, we now better explain constitutive agreement costs where simultaneously pursuing at multiple scales could dilute limited resources or political capital (lines 150-154).

Comment: I would also like to see the authors elaborate on the timescale, is there time to secure effective agreements (at any/all scales) with a sequencing approach in view of the how fast the 1.5 degree target window is closing?

Response: We have added a discussion paragraph elaborating this point. Specifically, our study focuses on changes in the relative rate of emission reductions, which is irrespective of the timescale. Because we model only interactions among states as a self-contained system that is separate from realized changes in climate (e.g., natural disasters take a probabilistic form and follow decades after changes in emissions), the time scale depends only on how often players are willing to re-evaluate and re-negotiate their climate policy. Absolute predictions of time scale would thus require a much more detailed model of both the negotiations process and climate dynamics, and thus is an important topic for separate, future research.

Comment: My main concern is thus that the paper has not reached its full potential in how relevant and useful it can be, through clear references to the current legal and policy regime. I also felt that the part on ecological restoration was separate from the main argument and should rather be discussed in a separate paper, to instead use the space to elaborate on the validity of the core assumptions behind climate change emission reduction, as suggested above.

Response: Thanks to the very helpful reviews, we feel the revised manuscript frames (lines 43-73) and discusses (lines 220-250) our study and results in current climate policy and theory to a much better degree. We removed the ecosystem restoration section.

Reviewer #3

Comment: This paper argues that global negotiations are slow and that regional ones can be faster but since a global participation is obviously needed, the authors argue in favour of a regime that mixes the two starting regional and at some point switching to global.

The argument of the paper starts by saying that “premature” global treaties don’t work. Well yes, it is true that nothing has worked yet. But is this due to lobbyism, rivalries related to other issue, a lack of understanding, faulty design, disagreement over burden sharing or what?

Response: Thank you for pointing out this lack of clarity! We rewrote the framing of our study to explicitly connect our model to the existing literature and policy landscape (lines 43-102). Specifically, we focus on legally-binding agreement strategies that, using sanctions, could incentivize stringent mitigation commitments (e.g., in line with a <2C warming scenario) to complement rather than replace the (voluntary) emissions reductions under current agreements (e.g., Paris, USA-China).

Our key finding is that combining an early emphasis on local legally-binding (i.e., sanction-backed) agreements with a timely shift to renewed negotiations for a stringent, legally binding global mitigation global treaty provides the fastest path to deepening current mitigation commitments. Our original and newly added analyses (discussed below) show that this approach can overcome several cornerstone challenges: expense of early technologies, economic rivalry, the disproportionate diplomatic influence of the largest emitters, and the high costs of a stringent mitigation policy, which we take to implicitly include many other unknown and unforeseeable barriers such as lobbying, faulty design, and differing perceptions about burden sharing.

Comment: The paper makes some interesting points and is worth a read but did not fully convince me. It is clear that global climate negotiations face problems but is this the solution? There are some regions – notably the EU where there has been some progress in climate policy but there are many regions where one fails to see much progress and – worse than that one fails to see why there would be any particular compelling logic to the regional approach.

The paper continues to argue that regional treaties will work much better. This is possible but I am not entirely convinced of the arguments. Of course if we accept this idea then it would be a good idea to start regionally and the second part follows automatically : one has to switch at some point from the regional to the global.

Response: We apologize that our central arguments were unclear in the submitted version, and we have clarified them in the revision. Overall, we are not arguing that regional treaties will work better, per se, but rather we are showing a sanctioning framework explored in the common pool resource literature, and which has analogs to many aspects of climate negotiations, implies the “switching/shifting” negotiation approach we explore here. We now also point out that the incentive structure in our models is similar to that of sanction-based climate clubs (Nordhaus 2015) but where for local agreements (unlike global agreements) clubs only admit players in their group and only punish non-mitigators within their group. Again, we now emphasize that the sanction-backed agreements we propose are intended to complement rather than replace voluntary commitments in existing agreements (e.g., Paris, US-China).

In our revised Introduction and Discussion, we now better explain that local agreements capitalize on chance events in individual player decisions, including incentives such as extreme climate events not modeled here explicitly. Such local events carry greater weight within the local groups than world-wide and allow mitigation to arise in some groups before others. Local agreements are then critical to sustaining local mitigation commitments in the face of subsequent chance events that may de-incentivize mitigation (e.g., energy crises favoring coal power). Thus, local agreements lock-in mitigation in early-adopting regions, and gradually ‘ratchet up’ mitigation world-wide. Our expanded references point out that this advantage of local agreements was described in a previous building blocks paper (Vasconcelos et al 2013) and complements other benefits of local agreements found in Environmental Agreements studies.

We now better highlight our key insight: that as soon as local agreements envelop and sustain more countries in stringent mitigation commitments (i.e., following <2C warming scenarios), renewed pursuit of a (presently untenable) global, stringent and sanction-backed climate agreement – instead of continuing the local approach – can rapidly expand mitigation. In a supplementary analysis, we now illustrate how delayed shifts to such a global agreement (e.g. if the shift to global negotiations happens when 80% of countries commit to mitigation while only a 50% majority is needed to enact the global agreement) erode the benefits of a multi-faceted strategy.

In the discussion, we now also re-emphasize our focus on the benefits of combining local and global mitigation incentives rather than the specific sanction structure of our model, which tailors to developing policy theory and intuition rather than specific policy solutions (lines 239-250). We discuss several potential alternative incentives at both local and global levels, and point out that our focus is on a ‘worst-case’ scenario where climate negotiations (e.g., USA-China talks) face severe barriers (e.g., rivalry) and without added incentives (e.g., tariffs) concerns of climate change damages in the largest emitters underlie few policy changes (e.g., as projected damages focus on lower-emitting, developing countries) (lines 220-238). Finally, we point how our model can be developed to provide specific guidance on future negotiations (e.g., timing, trigger threshold for shift to global agreements) and how they can alter climate trajectories (lines 239-250).

Comment: A very reasonable alternative approach is to say that you need a G2 or G3, G4 approach. The two biggest emitters are the US and China. Without them – no deal. One could argue that a reasonable approach is for these two mega-polluters and mega-powers to negotiate either bilaterally or preferably perhaps including at least the EU and India. This is the approach taken by the debate on “Climate Clubs” but the paper makes no reference to this debate nor Bill Nordhaus and others who have proposed such solutions.

Response: We revised the Introduction to emphasize that the incentive structure in our models is similar to the climate clubs approach (i.e., once sufficiently numerous, mitigators pass agreements to punish non-mitigators), but with an explicit consideration of scale: potential club members (and the players punished by the club) are either limited to a local group or region (local agreements) or drawn from across the world (global agreements, as in Nordhaus 2015). We also developed our model structure to show that G2/G3/G4 summits amongst the largest emitters to reach legally-binding mitigation commitments that go beyond Paris pledges represent one potential approach to local agreements (Figure 4). In fact, a world-wide emphasis on local legally-binding agreements in this scenario could also allow lower-emitting players to form legally-binding agreements that advance emissions reductions even in the event that the largest emitters remain locked in negotiations.

Comment: The paper does also not properly relate to the work by Scott Barrett on how to build coalitions and design treaties that will last and create incentives to deal with the incentive

problems involved. A third approach is to break up the global treaties and deal with different pollutants, sectors or industries separately. Ie to have one treaty for HCFCs and PFCs and one on deforestation and land use change and another on steel and cement etc. None of this, and very little in general of the economic or social science literature is referenced or used. Instead the authors focus exclusively on the regional vs global dimension.

Response: We have revised the Discussion to better emphasize that our results highlight the benefits of shifting between (or coordinating) local and global mitigation incentives and that, in practice, the formation and incentive structure of these local and global agreements can vary (lines lines 220-238). In addition to sanction-based climate clubs which most closely resemble the mechanisms in our model, we mention several of these alternative (or complementary) approaches, including International Environmental Agreements (following Barrett 1994 and more recent work) and regime complex approaches. Finally, our revised model framing focuses on local groups rather than geographic regions per se, where groups form around shared economic, political, or climate mitigation goals, and could vary in size (Figure 4).

Comment: The paper is backed up with a model about this transition that I found somewhat opaque and somewhat construed. It is quite mechanical – I don't see the economic forces, the interests, the lobbyists. Not much effort is made to motivate model structure or parameters. Somehow I was, again, not very convinced. Are the regions equal in size? What are the real driving forces?

Response: Thank you for raising this issue. We now explicitly link the modeled processes to existing theory on climate clubs (lines 93-99), highlight insights from our theoretical model inform other negotiation frameworks (lines 220-238), and greatly improve the clarity of the Methods.

Our revision also clarifies to a much better extent that our focus is on what combination of local and global legally binding agreements can best deepen current mitigation commitments under, voluntary pledges. Given this qualitative question, we intentionally begin with a simple model to better understand why building up agreement scales accelerates mitigation more than using only local or only global agreements. Subsequently we now extend this model to verify that this result holds for a greater range of more realistic scenarios (Figure 4). On lines 239-250 we highlight key mechanisms mentioned by the reviewers that are unlikely to affect our qualitative result, but accounting for which is key in developing our core finding into a potential negotiation framework (e.g., one where economic forces and enforcement mechanisms behind local agreements differ among regions).

Comment: All in all the paper makes some good points and is interesting but I fail to be totally convinced and I also see a lack of connection to the broader literature on treaty formation. In summary I am hesitant to recommend publication at least not without a very substantial revision.

Response: Thanks to the very helpful reviews, we feel the revised manuscript better highlights our key results, and frames (lines 43-102) and discusses (lines 220-250) our study and results in current climate policy and theory to a much better degree.

More detailed comments and questions

Comment: (p. 4, r. 83-84) How is this seen in figure 1? The variance of the local agreement is so high. It is not obvious that we are speaking of a doubling here. Please explain.

Response: Thanks for pointing this out; we have corrected this sentence to note that the improvement in rate is two-fold but the increase in total mitigation is less than that. As Fig. 1 focuses on illustrating the dynamics, we refer to Fig. 2 for an aggregate comparison across many (100) replicate simulations.

Comment: (p.4, r. 84-85) It is difficult to see in the figure what is claimed in the text that the figure shows. Doubling the speed to 80%? How does one see that in the figure?

(p.4, r. 91-92) I do not understand how one sees this in the figure. When we speak of a majority of countries – is this assuming they are identical – or do you mean a set of countries with a majority of the pollution – or of GDP?

Response: Thank you, we modified Figure 1-2 legends to emphasize that these analyses assume equal emissions levels across countries (players), such that “majority” corresponds to >0.5 and 80% corresponds to 0.8 in Fig. 1a. We now clarify that in the base model player emissions are homogeneous (i.e., “majority” refers to proportion of countries) except for the newly added subsection where we explore differences in emissions and, simultaneously, in diplomatic influence on establishing agreements.

Comment: (p. 7, r. 138-142) I don't understand this. Could you elaborate?

Response: Thank you, we rewrote this paragraph.

Comment: (p. 11, r. 228) More explanation needed, What is p_T ? How do we go from expression in row 226 to expression in row 228? Some more explanation would be beneficial at least in appendix or for the referee.

Response: Thanks for catching this; we now better explain the shifting function and that p_T denotes the quorum (fraction of players with mitigating strategy) at which agreements to punish non-mitigators are established.

Comment: (p.11, r. 230) Please show how you arrive at eq 2 and eq 3?

Comment: (p.11, r. 231) How do you motivate this feature of a random switch in strategy? What real world concept does it aim to capture?

Response: Thank you, we rewrote the Methods to be clearer and better-organized. In particular, we now explain the components of each expression immediately before it is presented.

Comment: (p. 10, r. 208-211) Are the thresholds really correct here? Hard to follow. If the players focus ALL their efforts on local agreements up until T_L , then what significance does T_G have? Since T_G happens before T_L ... Please explain and motivate better.

(p. 11, r. 224-225) For $T=0$ and $b>1$ you say the global and local agreements overlap. But previously (p.10, r. 213) you said $T=0$ means only global agreements will take place. This is somewhat confusing.

Response: Thanks for pointing out our indeed confusing formulation. We simplified the model to a single shift threshold T , where now with agreement overlap, $T=0.5$ and local agreements are discontinued at $T+b$ after global agreement attempts begin at $T-b$ (where now $b/2$ is the proportion of global mitigation levels for which local and global agreement attempts co-occur).

Comment: (p.12, r. 239-240) Is strength of rivalry is randomized?

Response: Yes, we specified that these terms are randomized and drawn from a multivariate uniform distribution, where there is a large degree to which the competitive effect of group A on B is correlated to the effect of group B on A (i.e., reciprocated competition over shared resources, markets, etc).

REVIEWERS' COMMENTS

Reviewer #1 (Remarks to the Author):

The revision has improved the manuscript a lot. Especially the revised Introduction, Discussion and Methods section clarify things for the reader. I also appreciate the clear focus on climate change (removal of the ecosystem restoration part) in the revised version.

However, I still suggest to improve the intuition in the description of the results (lines 103-189). The formal description of the model is in the methods part but it would help the reader to be more explicit about the main driving forces behind the findings (without any need to be formal here). Examples for this include:

Lines 116-119: What does "the role of payoffs in player climate policy choice" mean? Why are the mentioned aspects increasing the benefits of a shifting-regime?

Line 121: What are "benefits of among-group sanctions"? This is confusing because sanctions are costly, both for non-mitigators and for mitigators.

Lines 144f.: "the positive feedback by which group rivalry holds back mitigation arises at the global scale." This should be explained, what does positive mean here?

Line 158: What does "disproportionally" mean here?

Further minor comments:

Line 225: "Vasconcelos et al. (2013)" is the only time that a reference is given by author-year.

Line 330: First word should be players instead of player?

Line 308: in lieu.

Reviewer #3 (Remarks to the Author):

I am satisfied with the authors work to take my suggestions into account. I now recommend publication.

Reviewer #1:

Comment: The revision has improved the manuscript a lot. Especially the revised Introduction, Discussion and Methods section clarify things for the reader. I also appreciate the clear focus on climate change (removal of the ecosystem restoration part) in the revised version.

I still suggest to improve the intuition in the description of the results (lines 103-189). The formal description of the model is in the methods part but it would help the reader to be more explicit about the main driving forces behind the findings (without any need to be formal here). Examples for this include:

Response: we thank the reviewer for a second positive and thorough review of our work, and detail our corrections below.

Lines 116-119: What does “the role of payoffs in player climate policy choice” mean? Why are the mentioned aspects increasing the benefits of a shifting-regime?

Response: Thank you, we re-wrote and expanded ideas in this sentence (now lines 119-124). We also expanded the introduction of the model (lines 81-85) to clearly explain the process of player strategy choice here as well as in the Methods.

Comment: Line 121: What are “benefits of among-group sanctions”? This is confusing because sanctions are costly, both for non-mitigators and for mitigators.

Response: Thank you, we replaced this sentence with a clearer explanation of the idea on lines 116-119. The “benefit” we previously referred to is stated earlier in the paragraph (“sanctions in global agreements push non-mitigating groups to cooperate on reducing emissions”).

Comment: Lines 144f.: “the positive feedback by which group rivalry holds back mitigation arises at the global scale.” This should be explained, what does positive mean here?

Response: Thank you, we rewrote and better explained this section (lines 147-152).

Line 158: What does “disproportionally” mean here?

Response: Thank you, we rewrote and better explained this sentence.

Comment: Further minor comments:

Line 225: “Vasconcelos et al. (2013)” is the only time that a reference is given by author-year.

Line 330: First word should be players instead of player?

Line 308: in lieu.

Response: Thank you for catching these, we made the changes.

Reviewer #3:

I am satisfied with the authors work to take my suggestions into account. I now recommend publication.